# Immuno-Oncology at the Crossroads: Confronting Challenges in the Quest for Effective Cancer Therapies

**DOI:** 10.3390/ijms26136177

**Published:** 2025-06-26

**Authors:** Claudiu Natanael Roznovan, Luminița Gabriela Măruțescu, Gratiela Gradisteanu Pircalabioru

**Affiliations:** 1Department of Botany and Microbiology, Faculty of Biology, University of Bucharest, 030018 Bucharest, Romania; claudiu-natanael.roznovan@s.unibuc.ro (C.N.R.); gratiela.gradisteanu@icub.unibuc.ro (G.G.P.); 2Research Institute of University of Bucharest, 050107 Bucharest, Romania; 3eBio-Hub Centre of Excellence in Bioengineering, National University of Science and Technology Politehnica Bucharest, 060042 Bucharest, Romania

**Keywords:** immuno-oncology, immunotherapy, immune checkpoint inhibitors, tumor microenvironment, tumor–immune interactions, microbiome, precision oncology, biomarker-guided treatment

## Abstract

Immuno-oncology has rapidly evolved into a cornerstone of modern cancer therapy, offering promising avenues for durable responses and personalized treatment strategies. This narrative review provides a thorough overview of the mechanisms underlying tumor–immune system interactions and the therapeutic innovations emerging from this knowledge. Central to this discussion is the tumor microenvironment (TME), a complex ecosystem of immune and stromal cells that supports tumor growth and shapes therapeutic outcomes. Key cellular and molecular factors within the TME are examined, along with diverse immune escape strategies. We further analyze the landscape of immunotherapeutic approaches, including immune checkpoint inhibitors, cancer vaccines, adoptive cell therapies such as CAR-T cells, and cytokine-based interventions. This review also addresses the increasing importance of predictive biomarkers in immuno-oncology, particularly in patient stratification, monitoring resistance, and managing immunotherapy-related toxicity. Finally, we explore the emerging role of the microbiome as a modulator of immunotherapy efficacy, shedding light on host–microbe–immune interactions that may influence clinical outcomes. By integrating current biological insights with therapeutic innovation, this review outlines the challenges and opportunities ahead in immuno-oncology and emphasizes the need for translational research and cross-disciplinary collaboration to optimize cancer immunotherapy in the era of precision medicine.

## 1. Introduction

Cancer remains a significant and persistent threat to global public health, with a rising incidence and significant mortality burden despite major advancements in therapeutic strategies [1,2]. Traditional modalities such as surgery, chemotherapy, and radiotherapy, although effective in certain localized tumors, often fall short in metastatic or resistant disease contexts [3,4]. The advent of immunotherapy, particularly immune checkpoint inhibitors (ICIs) and personalized cancer vaccines, has revolutionized oncologic treatment paradigms by leveraging the immune system’s mechanisms to recognize and destroy malignant cells [5,6,7]. However, the effectiveness of these strategies varies considerably across different patient cohorts and tumor types, highlighting the critical need for reliable predictive markers and a more comprehensive understanding of tumor–immune system dynamics [8].

A crucial determinant of response to immunotherapy lies within the tumor microenvironment (TME), a complex, dynamic milieu composed of immune cells, fibroblasts, vasculature, and extracellular matrix components [9,10,11]. In recent years, research has pointed out the significant role of non-malignant stromal elements, including tumor-associated macrophages (TAMs), cancer-associated fibroblasts (CAFs), and tertiary lymphoid structures (TLSs), in shaping immunotherapeutic outcomes [11,12]. Emerging concepts such as antigen-presenting CAFs and immune-activating TLSs represent promising new biomarkers and therapeutic targets [12,13].

This review distinguishes itself by integrating recent findings across multiple axes of cancer immunotherapy, highlighting novel cellular contributors within the TME, such as antigen-presenting CAFs and CD8+CD103+ tissue-resident memory T cells [14], and exploring state-of-the-art technological platforms for spatial and molecular characterization. Notably, we delve into underexplored elements such as the predictive value of the gut microbiome [15] and tertiary lymphoid structures in shaping immune response and treatment efficacy. Furthermore, this review contextualizes the epigenetic regulation of immune evasion and therapy resistance, a growing frontier in cancer immunology. The aims of this review are multifaceted, reflecting the complexity and evolving landscape of cancer immunotherapy. First, it seeks to dissect the role of components within the TME, especially novel immune and stromal cell subsets such as antigen-presenting fibroblasts and tissue-resident memory T cells, in modulating the response to these therapies. Secondly, we summarize the latest advances in immunotherapeutic strategies, with a particular focus on ICIs, cancer vaccines, and adoptive cell transfer approaches. Thirdly, this review aims to evaluate both current and emerging biomarkers for predicting therapeutic efficacy and toxicity, with a particular emphasis on spatially resolved imaging and multi-omics platforms. In addition, it explores the mechanisms underlying immune evasion, including those driven by epigenetic alterations, and proposes integrated strategies to overcome resistance. Finally, this review highlights the translational and clinical implications of these findings, offering a forward-looking perspective that supports the advancement of precision immuno-oncology.

## 2. Immune Evasion Mechanisms in Cancer

Cancer cell survival is strictly dependent on its ability to bypass immune detection. Immunosurveillance acts as the primary selective pressure and is therefore a key mechanism through which negative selection drives the clonal expansion of the cells that have escaped immune control. Immune evasion, accordingly, represents the hallmark of cancer, and the mechanisms that lead to it are a critical topic of discussion [16,17,18], often representing mutations that have enabled cancer cells to, among others, take advantage of peripheral immune tolerance pathways [19,20], modify antigen presentation [21], or lose their antigenicity altogether [22].

### 2.1. Mechanisms Directly Involving T Cells

T cells are a centerpiece in the complex interplay between normal stromal cells, cancer cells, and immune effectors, within the TME. Naturally, evading T-cell response is crucial for disturbing the state of equilibrium or senescence in which the immune system maintains tumors in a dormant state [22]. As previously mentioned, physiological immune tolerance pathways can serve as gateways for suitably mutated cancer cells. These pathways rely on controlled T-cell infiltration and activation via antigen recognition [16]. Cancer cells’ control over T-cell trafficking is realized through multiple mechanisms.

Endothelial adhesion is a crucial step in T-cell recruitment; consequently, endothelial dysfunction in tumor-associated endothelial cells contributes to T cells’ exclusion from the TME. This exclusion may result from the downregulation of adhesion molecules (ICAM-1, ICAM-2, E-selectin, VCAM-1) driven by tumor-derived pro-angiogenic factors such as vascular endothelial growth factor (VEGF), which is upregulated in response to hypoxic conditions within the TME [23,24]. VEGF also induces the expression of the Fas ligand (FasL) and PD-1 ligand (PD-L1) on the surface of the tumor endothelial cells, promoting T-cell apoptosis and thus further aggravating T-cell exclusion [25]. 

Another mechanism involves the restriction of T cells to the stromal areas, preventing their direct contact with cancer cells [26]. Increased TGF-β expression in tumors plays an key role in this process by inducing the formation of cancer-associated fibroblasts (CAFs) [27]. CAF-mediated T cell arrest occurs either indirectly through the formation of collagen-rich physical barriers [28] or directly via antigen cross-presentation, FasL-mediated CD8+ T cell killing, and the expression of CXCL12, which exert a chemorepellent effect on CXCR4 expressing T cells [29]. 

RNA sequencing has revealed a high heterogenicity of CAFs populations, correlating with contrasting clinical outcomes following their depletion. While the immune evasion mechanisms of specific CAF subtypes remain largely unknown [30,31], Chakravarthy et al. recently identified a subset of ECM genes, specifically dysregulated in cancer, that are modulated in CAFs through a transcriptional program associated with TGF-β signaling in the TME and its enrichment in hot tumors, indicating a function in immune escape. This ECM-based immune evasion signature outperforms traditional biomarkers such as tumor mutation burden, cytolytic activity, TGF-β expression, CAF abundance, and T-cell-inflamed signatures in predicting immune checkpoint blockade response. Notably, this signature holds promise both a therapeutic target and as a stratification tool for stratifying for precision immunotherapy [32].

The differential production of chemokines involved in T-cell recruitment is evident in comparison between T-cell-infiltrated tumors and an immune cold TME. Among them, CXCR3 ligands such as CXCL9, CXCL10, and CXCL11, produced consequently to IFN-γ signaling, have been extensively studied [33]. Both epigenetic and non-epigenetic alterations can be responsible for this. The hypermethylation of receptor/chemokine promoter caused by the activity of complexes such as polycomb repressive complex 2 (PRC2) [34] or mutations in (1) ErbB family epidermal growth factor receptor (EGFR) [35] and (2) peroxisome proliferator-activated/retinoid X receptor (PPARg/RXRa) (both leading to decreased levels of CCL5 and CXCL10 and consequently limited T-cell infiltration) [36], respectively, are a few examples.

In the context of altered antigen presentation, deficient MHC class I expression by cancer cells facilitates their escape from T-cell mediated immune responses. This deficiency can result from mutations or epigenetic alterations in the structural components of MHC class I molecules (such as heavy chains, β2-microglobulin), other antigen processing machinery (APM) components (TAP molecules, tapasin), or in transcription factors regulating MCH expression such as NRLC5 [37]. Additionally, reduced expression of IFN-γ, although not always [38], and NF-κB [39] also lead to decreased MCH class I expression.

The loss of neoantigens or tumor-associated antigens under immune pressure has been observed, especially in passenger mutations - by-products of tumoral proliferation that are not crucial for tumorigenesis and can thus be discarded. Antigenic loss may occur through multiple mechanisms, including genomic copy number loss, transcriptional downregulation via epigenetic modifications, or post-translational alterations [40]. Additionally, reduced neoantigen expression has been reported in the context of copy number-independent, allele-specific gene expression, although the underlying mechanisms remain poorly understood [41].

### 2.2. Mechanisms Involving Other Immune Cells

NK cells are key immune effectors in the anti-tumoral response. Under physiological conditions, MHC class I molecules—and the consequent activation of T cell—suppresses NK cell activity through interactions with inhibitory killer cell immunoglobulin-like receptor (KIRi) and via HLA-E-NKG2A signaling, triggered by HLA-1 peptide. In contrast, when MHC class I expression is downregulated, as often observed in the TME, NK cells are activated through the “missing self” recognition mechanism. However, under sustained NK cell-mediated immune pressure, tumor cells can develop evasion strategies that are bypass the classical HLA-I pathways. These include the upregulation of non-classical HLA-E molecules, both on tumor cells and antigen-presenting cells (dendritic cells and macrophages) the impairment of activating receptors such as NKG2D, and the induction of inhibitory immune checkpoints such as PD-1 [11]. Conventional dendritic cells (cDC1) are crucial in tumoral antigen presentation and thus T-cell priming, therefore the inhibition of cDC1 recruitment into the TME represents another immune evasion mechanism. This is achieved through reduced secretion of CCL4/CCL5 by cancer cells, a process driven by activation of the β-catenin signaling pathway [42].

Tumor-associated macrophages (TAMs) are instrumental in tumors’ proliferation and immune evasion. The polarization of resident and circulating macrophages into TAMs is partially driven by soluble molecules secreted by tumoral cells such as sonic hedgehog (SHH) and succinate which promote M2 polarization. M2-polarized macrophages exert immunosuppressive effects, including inhibition of CXCL9 and CXCL10 leading to reduced CD8+ T-cell recruitment in the TME [43]. TAMs can also secrete VEGF which contributes to immune evasion as previously described [44].

Targeting the immune evasion mechanisms employed by cancer cells has emerged as a powerful strategy in cancer therapy, with modern immunotherapy approaches representing the most practical and impactful applications of these principles. 

## 3. Immunotherapy Approaches

The advent of cancer immunotherapy represents a paradigm shift in oncologic treatment strategies, surpassing classical approaches such as chemotherapy and radiotherapy in terms of both patient outcomes and adverse effects [45]. Stimulating the immune response and restoring immune control within the TME are two major principles guiding the development of cancer immunotherapies [46]. Currently, four such therapies that focus on boosting immune response within the TME have been approved by the FDA based on positive clinical trial results in cancer patients (Figure 1) [47].

### 3.1. Immune Checkpoint Inhibitors (ICIs)

Immune cell activation is mediated through receptor–ligand interactions, either between immune cells themselves or between immune and tumor cells. Specifically, the activation of tumor-specific T cells depends on the recognition of tumor-derived neoantigens by the T-cell receptor (TCR) and is further modulated by costimulatory and coinhibitory signals delivered through immune checkpoint molecules [48]. From a physiological perspective, immune checkpoints are essential for immunosurveillance and the resolution of autoimmune responses. However suitably mutated tumor cells can escape immune surveillance by developing mechanisms that disrupt the costimulatory–coinhibitory balance [49]. CTLA-4/B7-1, PD-1, and PD1-L1 represent the main immune checkpoints that inhibit T cell function and are targeted by currently approved ICIs. While their expression has been found to hold predictive value for treatment response, additional biomarkers are necessary to optimize efficiency and minimize toxicity across patient populations [50]. Currently FDA-approved monoclonal antibodies (mAbs) include ipilimumab (anti-CTLA-4), cemiplimab, pembrolizumab, nivolumab (anti-PD-1), durvalumab, avelumab, and atezolizumab (anti-PD1-L1). Response rates of monotherapy with anti-PD-1 mAbs range between 15 and 30%. Although combining it with anti-CTLA-4 agents improves response rates, it also increases toxicity. A promising strategy to address this involves combining single-agent ICI with personalized neoantigen-based tumor-specific vaccines, which prime the host’s defense mechanisms prior ICI therapy administration, thereby enhancing therapeutic efficacy [51].

### 3.2. Cancer Vaccines

Cancer vaccines have been developed as a strategy to restore T-cell infiltration (hot condition) in an immune cold TME, counteracting immune evasion mechanisms such as neoantigen silencing [52]. Unlike hematologic malignancies, where a common neoantigen is consistently expressed across malignant cells and can be effectively targeted, solid tumors often lose such antigens under therapeutic pressure or lack them entirely. Traditional cancer vaccines have shown limited success due to poor antigen selection, low immunogenicity, and inadequate patient selection. In contrast, modern vaccines efficacy relies on computational methods of identifying personalized, highly immunogenic neoantigens in real time. This process involves whole-exome sequencing to detect mutations and predict neo-epitopes with a high affinity for the patient’s MHC molecules [53].

Antigens selected for cancer vaccines should ideally meet the following criteria: (1) be constitutively expressed on tumor cells; (2) occur across multiple cancer types; (3) be essential to tumor survival; and (4) elicit a robust immune response. Currently, the two major categories of antigens are considered promising candidates: (1) tumor-associated antigens (TAAs) which are abnormally expressed self-proteins [54] including surface oncogenes (HER2) [55], cell lineage differentiation antigens (PSA) [56], or other mutations (BCR-ABL1) [57]; and (2) oncogenic viral antigens (HBV, HPV etc.), found in virus-driven cancers, which account for approximatively16% of all cancers [58].

Another important aspect influencing the success of neoantigen vaccines is the delivery platform, which is closely linked to tumor mutational burden (TMB). More specifically, a key challenge lies in effectively delivering vaccine components to specific sites, such as secondary lymphoid organs near the tumor, or directly into the TME. Antigens can be administrated directly or loaded into DCs [59]. Direct administration involves injecting a source of antigen via various routes (intramuscular, intravenous, subcutaneous, or intracutaneous) and therefore introducing them into the antigen-processing pathway of DCs. Promising delivery formats include DNA, RNA, and synthetic long peptides (SLPs) [60,61].

DNA vaccines are relatively easy to produce; however, require transcription and translation before cross-presentation by DCs [62]. Their ability to activateCD4+ and CD8+ T-cell responses is most effective following high-dose intramuscular administration [63]. A DNA vaccine targeting the E6 and E7 oncogenes of HPV-16 and HPV-18 has demonstrated efficacy in treatinghigh-grade intraepithelial neoplasia [64].

RNA vaccines share many advantageswith DNA vaccines, except for the extra need of transcription, being closer to antigen expression, processing, and presentation [65]. Administration routes explored include direct injection into lymph nodes [66] or intravenously via lipid nanoparticles [67]. Encouraging results has been observed in melanoma patients who received an intranodal mRNA vaccine encoding ten specific neoantigens, resulting in a strong vaccine-specific antitumor immune response and a significant reduction in metastatic events [66]. Intravenous delivery of RNA-loaded lipoplex particles has been shown to selectively target DCs in the spleen and lymph nodes, eliciting a more robust immune response than localized approaches, such as subcutaneous, intracutaneous, or intramuscular injections. Clinical trials have demonstrated that RNA vaccines can break immune tolerance to tumor-associated antigens and enhance the efficacy of complementary treatments [58].

### 3.3. Synthetic Long Peptides (SLPs)

SLP vaccines were developed to overcome the suboptimal response observed with early peptide vaccines that used MHC-I-binding short peptides. While these peptides are presented by MHC-I molecules on all nucleated cells, only DCs can provide the necessary co-stimulatory signals for effective T-cell activation. As a result, early vaccines often lead to weak immune responses and depletion via anergy. SLP vaccines require processing by professional antigen-presenting cells, resulting in DC-centered antigen presentation within the draining lymph nodes and optimal MHC-I presentation [68]. SLP vaccines formulated in incomplete Freund’s adjuvant (IFA) avoid the T-cell accumulation and depletion seen in short MHC-I-binding peptide vaccines, leading to more robust and durable immune responses. Recent trials suggest that the incorporating adjuvants such as TLR ligands and CD4+ T cell helper peptides can further enhance immune activation, emphasizing the importance of the adjuvant selection and CD4+ T-cell support in generating effective CD8+ responses. Clinical studies have demonstrated the efficacy of SLP vaccines in both premalignant and malignant HPV-16-induced lesions [69].

DC-based delivery involves isolating DCs, activating them with adjuvants, and loading them with antigens using various techniques including direct pulsing with neoantigens, mRNA electroporation, lentiviral transduction, fusion with cancer cells, or co-culture with whole-tumor lysate [70]. Administration routes include intracutaneous, subcutaneous, or intravenous injection. In murine studies, injected DCs, not directly prime T cells, but also deliver antigens to the body’s own cross-presenting DCs. In most clinical trials, monocyte-derived DCs are used due to the difficulty of obtaining more functionally specialized subsets in sufficient quantities. However, these monocyte-derived cells lack a complete set of co-stimulatory molecules and have a limited cross-presentation capacity. Different DC subsets have distinct roles: conventional DC1 (cDC1) are particularly effective at cross-presenting antigens to CD8^+^ T cells, while cDC2 are more efficient at priming CD4+ T-cell. It remains unclear how using various subsets of DCs, alone or in combination, affects vaccine efficacy. Advances in culture systems now allow large-scale generation of specific DC subsets from umbilical cord blood and peripheral blood derived stem cells, potentially enhancing the effectiveness of DC-based cancer vaccines [71,72,73]. Sipuleucel-T is the only FDA-approved therapeutic cancer vaccine. It is a DC-based immunotherapy approved over a decade ago and it is produced from peripheral blood mononuclear cells (PBMCs) collected via leukapheresis. DCs are activated ex vivo using a fusion protein composed of a prostate tumor antigen and granulocyte–macrophage colony-stimulating factor (GM-CSF). This vaccine has been shown to improve survival in patients with metastatic castration-resistant prostate cancer by approximately four months [74].

### 3.4. Adoptive Cell Transfer (ACT) Therapy

Adoptive cell transfer (ACT) is a form of cell-based immunotherapy that uses tumor-reactive immune cells to selectively detect and destroy tumor cells expressing specific tumor-associated antigens. T cells are the primary candidates for adoptive cell immunotherapy, with two main strategies currently employed in T-cell transfer approaches. The first involves the isolation of tumor-infiltrating lymphocytes (TILs) from immunologically active (“hot”) tumors, followed by their ex vivo proliferation using T-cell-stimulating cytokines such as interleukin-2 (IL-2). These expanded TILs are then reinfused into the patient, often in combination with adjuvants that enhance the antitumor activity. However, this approach is limited by the frequent absence or low number of TILs in patients who are immunosuppressed due to prior radiotherapy or chemotherapy [75]. In the second strategy, T cells are genetically engineered to express chimeric antigen receptors (CARs) that enable them to identify tumor cells expressing distinct surface markers. A typical CAR construct includes an extracellular single-chain variable fragment (scFv) that recognizes and binds the target antigen, linked to intracellular signaling domains derived from co-stimulatory molecules such as CD28 or CD137 (4-1BB). This design enables CAR-T cells to be activated independently of MHC presentation, allowing direct tumor recognition and cytotoxicity [76].

CAR-T-cell therapies targeting CD19-expressing cancer cells have demonstrated substantial clinical efficacy in eliminating malignant hematologic cells. Clinical trials have reported response rates ranging from 25% to 90% in diseases such as B-cell lymphoma, non-Hodgkin’s lymphoma, and chronic lymphocytic leukemia. As a result, in 2017, two CD19-directed CAR-T-cell therapies—tisagenlecleucel and axicabtagene ciloleucel—received FDA approval [77,78,79].

Due to challenges in lymphodepleted patients and the time-consuming process of autologous T cells gene transduction, alternative ACT strategies have explored the use of CAR-NK cells derived from human induced pluripotent stem cells (iPSCs). These cells can exert anti-tumor effects without requiring HLA matching and can be generated in unlimited quantities, offering a cost- and time-efficient therapeutic option [80].

### 3.5. Cytokine-Based Therapies

Within the TME, cytokines are key factors in supporting immune cell growth, activation, and infiltration. Notably, the administration of IL-2 has been shown to exert strong anti-tumor effects and the ability to inhibit the progression of metastatic tumors in murine models [81]. IL-2 therapies were authorized by the FDA in 1992 for renal cell carcinoma and melanoma treatment, with clinical r response rates ranging from 15% to 29%. IL-2 promotes T cell proliferation and activation, and high-dose IL-2 has shown the capacity to elicit potent immune responses against cancer cells. However, its use is associated with significant toxicity, including multi-organ complications, hypotension and capillary leak syndrome [82]. To mitigate these side effects, low-dose IL-2 combined with ACT has shown encouraging clinical results. Furthermore, newer ACT approaches utilize orthogonal IL-2 cytokine–receptor pairs that selectively target engineered receptors on CAR-T or CAR-NK cells, thereby minimizing off-target toxicity in healthy tissues [83].

### 3.6. Resistance to Immunotherapy

Immunotherapy resistance can be categorized based on the timing when it emerges and underlying mechanisms. Importantly, immunotherapy responses are dynamic and evolve throughout disease progression, influenced by. by both intrinsic physiological factors and extrinsic therapeutic interventions.

Primary immune resistance refers to patients who never respond to immunotherapy, often due to a lack of tumor antigens, which prevents T cell recognition. In contrast, some tumors develop adaptive resistance by modifying their microenvironment to evade immune detection by downregulating or concealing tumor antigens. effectively evading immune detection and disabling the adaptive immune response. Finally, some patients initially respond to immunotherapy, but later, relapse, sometimes with new metastatic lesions. This scenario is known as acquired (or secondary) immune resistance, and it reflects the tumor’s ability to evolve and escape immune control after an initial period of effectiveness [84].

Primary and adaptive resistance may result from intrinsic mechanisms that fall into one of the following four categories: (1) absence of tumor antigens and deficient T-cell detection (2) defects in antigen presentation and lack of HLA molecules, (3) constitutive expression of immune checkpoint ligands (4) dysregulation of signaling pathways, including MAPK, PI3K, WNT/β-catenin, and IFN-γ [85]. Or it can be the result of extrinsic mechanisms such as the following: (1) activation of immunosuppressive cells (e.g., Tregs, MDSCs, CAFs, M2 macrophages, and N2 neutrophils) often induced by TGF-β); (2) secretion of immunosuppressive factors (e.g., TGF-β, TNF-α) into the TME; and (3) increased expression of immune checkpoints like PD-1, CTLA-4, LAG3, TIGIT, and TIM-3 [86].

Although the mechanisms behind acquired resistance are not yet fully understood, they appear to share features with primary resistance, including a diminished T-cell functionality, antigen loss or alteration, impaired antigen recognition, an increased expression of immune blockers, and accumulation of immunosuppressive cells in the TME.

Resistance to immunotherapy remains a major barrier to the widespread success of immunotherapy. While the mechanisms of primary, acquired, and adaptive resistance (Table 1) overlap with those used by cancer cells to evade immune detection, further research is needed.

A recent study by Bailey et al. identified extrachromosomal DNA (ecDNA) in 17.1% of 14,778 cancer patients confirming prior data that ecDNA-driven tumors are less responsive to ICIs due to translational patterns suggestive of immunosuppression. Approximately 34% of ecDNA-containing tumors harbored immunomodulatory genes involved in suppressing immune effector functions, inhibiting leukocyte-mediated cytotoxicity, and downregulating lymphocyte activation. These effects were validated by comparing the T-cell fractions in tumors with both immunomodulatory genes and oncogenes alone [87]. This research opens new perspectives for identifying novel targets to overcome resistance and represents a promising direction for advancing cancer immunotherapy.

**Table 1 ijms-26-06177-t001:** Mechanisms of resistance to current immunotherapy approaches.

Immunotherapy Type	Examples	Mechanism of Resistance	References
Immune Checkpoint Inhibitors (ICIs)	Anti-PD-1 (e.g., nivolumab) Anti-PD-L1 (e.g., atezolizumab) Anti-CTLA-4 (e.g., ipilimumab)	Loss of tumor antigen expression Mutations in IFN-γ/JAK/STAT pathway (e.g., JAK1/2 mutations) Upregulation of alternative immune checkpoints (e.g., LAG-3, TIM-3, TIGIT) Immunosuppressive tumor microenvironment (e.g., Tregs, MDSCs, TAMs) Activation of WNT/β-catenin signaling leading to T-cell exclusion	[88,89,90]
Adoptive Cell Therapies (ACTs)	CAR-T cells, CAR-NK cells	Antigen loss or downregulation on tumor cells Immunosuppressive cytokines in the tumor microenvironment Physical barriers preventing T-cell infiltration Exhaustion of transferred T cells	[88]
Cancer Vaccines	Peptide-based vaccines, dendritic cell vaccines	Low immunogenicity of tumor antigens Tumor-induced immunosuppression Antigenic variation leading to immune escape	[88]
Cytokine Therapies	Interleukin-2 (IL-2), Interferon-alpha (IFN-α)	Activation of regulatory T cells leading to immunosuppression Systemic toxicity limiting therapeutic doses Short half-life requiring frequent administration	[88]

## 4. Biomarkers Associated with Efficacy of ICI Therapy

The biomarker-associated efficacy, resistance, or toxicity of ICI therapy remain active areas of research. Currently, the primary biomarkers of ICI response authorized by the FDA are tumor PD-L1 protein levels, tumor mutational burden (TMB), and microsatellite instability (MSI). These biomarkers have demonstrated utility in identifying cancer patients more likely to benefit from ICI therapy. While none of these biomarkers are absolute, they offer significant clinical value in guiding treatment decisions.

### 4.1. Programmed Death Ligand 1 (PD-L1)

PD-L1 expression is often dysregulated in tumors as a key mechanism of immune resistance. Specifically, it suppresses T-cell expansion and effector function, both of which are essential for effective immune-mediated tumor control. Currently, PD-L1 expression in tumor tissues, evaluated via immunohistochemistry (IHC), is utilized as a biomarker to determine eligibility for PD-1/PD-L1 targeting ICI [91]. Although PD-L1 expression has been associated with response in certain cancers, significant clinical benefit has also been reported in patients lacking PD-L1 expression. Several factors may explain this: (i) PD-L1 expression is spatially heterogeneous, varying across different tumor regions or between biopsy specimens; (ii) its predictive value is limited, as other elements of the TME also influence therapeutic response; (iii) variability in assay platforms and testing protocols can lead to inconsistent results; and [92,93]; (iv) tissue-based biomarkers are dynamic and may evolve during treatment.

### 4.2. Tumor Mutational Burden (TMB) and Microsatellite Instability (MSI)

Tumor mutational burden (TMB), defined as the number of DNA mutations per megabase (muts/Mb), has been linked with ICIs efficacy [94,95]. In cancers such as NSCLC and melanoma, high TMB correlates with improved responses and clinical outcomes [96]. However, its predictive value varies across cancer types. For example, in breast cancer, TMB has shown limited utility, and in recurrent glioblastoma, patients with low TMB have still demonstrated favorable responses to immunotherapy [97]. These findings suggest that TMB alone provides a limited, one-dimensional view of tumor immunogenicity. Similarly, high microsatellite instability (MSI-H) has been strongly associated with improved ICI responses across various solid tumors, leading to the first FDA-approved tissue-agnostic therapies guided by MSI status.

### 4.3. Tumor Microenvironment (TME)

The TME is a tumor-derived complex structure with a predictive significance for disease progression and treatment outcome, making it a valuable therapeutic target. It consists of cancer cells, host immune and stromal cells and extracellular matrix enriched with secreted molecules—including cytokines, chemokines, and enzymes—as well as blood vessels. Host cells range from lymphocytes, granulocytes, and macrophages to fibroblasts and endothelial cells. The TME structure and features can significantly impact the response to immunotherapy.

The immune landscape of tumors, characterized by the presence of TILs, including B cells, T cells, natural killer cells, and myeloid cell, has been extensively investigated in different malignancies. The predictive value of these immune cells varies depending on cancer type and the specific immunotherapy employed.

#### Tumor-Infiltrating Immune Cells

CD8+ T cells are the key anti-tumoral immune effectors, via cytolytic activity and targets in immunotherapy. Gaining immune function is consequent to activation, which in turn is dependent on the MHC class I presentation of surface antigens on cancer cells. Cytolytic activity is carried through via FasL expression and the degranulation of granzyme and perforin into the immunological synapse with the tumor cell [98]. Cytotoxic T cells (CTLs) are polarized in different subsets; among them, CTL1 presents a potent cytotoxic activity and IFN-γ production, while CTLs produce less IFN-γ but retain their cytotoxic activity. On the other hand, CTL9 and CTL17 present low levels of granzyme B and therefore lack cytotoxic activity [99]. CTLs’ polarization can be driven towards the CTL17 and CTLs subsets, as a consequence of the activity of tumor-associated macrophages (TAMs) or during antigen cross-presentation by dendritic cells, respectively. The exact mechanisms by which the TME influences Tc polarization are still unknown, but CTLs’ subset composition can serve as a response indicator in immunotherapies like ICIs.

Following sustained T-cell receptor stimulation within the TME, CD8^+^ T cells progressively differentiate into a dysfunctional phenotype known as T-cell exhaustion [100]. The exhausted CD8+ TILs have impaired cytotoxic effector activity and an upregulation of inhibitory receptors, such as PD1, CTLA-4, LAG3, and TIM3, and increased levels of transcription factors TOX and TOX2 [101]. The goal of immunotherapy, and in particular ICIs, is to reverse T exhaustion in cancer. Unfortunately, the therapeutic use of ICIs is limited due to suboptimal responses in some patients, a low tumor penetration, toxicity, and substantial production costs. The mechanisms underlying resistance to ICI therapy remain poorly understood. It is unclear whether this resistance stems from an impaired reactivation of CD8^+^ T cells within the TME, insufficient peripheral recruitment of naïve CD8^+^ T cells from the periphery, or a combination of both. Addressing this uncertainty requires a more comprehensive characterization of the functional and transcriptional consequences of PD-1 expression in tumor-infiltrating CD8^+^ T cells.

A key predictor of the response to ICIs is TILs infiltration, including their density, phenotypic profile, and clonal diversity. A high TILs density has been identified as a strong predictor in several types of malignancies. However, it is not just the quantity of TILs that matters; their subtype and function (e.g., CTLs versus regulatory T cells (Tregs)), and spatial localization within the TME, are equally important. These factors influence whether TILs contribute to tumors’ control or progression. Notably, studies have shown that TILs’ density at the invasive margin, as measured by IHC, is more significantly associated with an anti-PD-1 therapeutic response than central tumor infiltration. An increase in TILs has been associated with a positive response to ICI therapy and positive prognosis in breast cancer [102,103,104,105], melanoma [106], colon cancer [107,108,109,110,111,112], NSCLC [113,114,115], gastric cancer [116], laryngeal squamous cell carcinoma [117], head and neck squamous cell carcinoma [118], vulvar cancer [119], hepatocellular carcinoma [120], and epithelial ovarian cancer [121].

Hashemi et al. [122] reported a clinical study that included a total of 366 patients with advanced NSCLC who received either nivolumab or pembrolizumab. The analysis included assessments of tumor and stroma for CD8+ T-cell infiltration. In patients with metastatic NSCLC, the stromal CD8 + TILs were identified as the strongest biomarkers for an improved outcome. Also, in another study, stromal TILs were identified as a biomarker for ICIs’ efficacy in HER2-positive breast cancer patients [123,124].

Across multiple cancer types, including melanoma, glioma, and NSCLC, as well as ovarian, bladder, and breast cancers [125], high intra-tumoral densities of CD8+CD103+ TILs were also identified as predictive biomarkers for ICI response, therapeutic cancer vaccines, and clinical outcome. The ICI therapy was demonstrated to trigger and increase the effector functions of CD103+CD8+ TILs against tumor cells [126]. These results indicate that CD8+CD103+ TILs have the potential to serve as biomarkers for ICIs response. Tumor-infiltrating Tfc cells are an additional potential biomarker that was identified to help select patients most likely to respond to ICIs, and they serve as biomarkers for clinical outcomes in high-grade serous ovarian cancer patients [127].

The evaluation of TILs in the TME is promising, but it remains costly and lengthy, and its integration into regular medical practice is impending [128]. Recently, a deep learning-based tool, named TILScout, was developed for the automated classification of whole-slide image patches and the calculation of TIL scores. The TIL scores were validated across multiple cancers, showing potential as prognostic and predictive biomarkers for antitumor response. Associations with cancer driver gene mutations suggest therapeutic relevance, though further clinical validation is needed [129]. In addition, an artificial intelligence-powered immune phenotype based on spatial TILs analysis was able to predict the ICIs’ efficacy in patients with biliary tract cancer [130], high-grade serous ovarian carcinoma [121], and lung adenocarcinoma [115].

The ICIs outcome is impacted not only by the localization and abundance of TILs but also their effector status. An increased immune cytolytic activity measured based on the expression level of granzyme and perforin 1 genes was associated with ICI monotherapy response in different types of cancers, including melanoma, hepatocellular carcinoma [131], gastric cancer [132], colorectal cancer [133], and lung adenocarcinoma [134]. Cytolytic activity can be evaluated using various methods, including TIL analysis via flow cytometry, gene expression profiling, imaging mass cytometry, and multiplex immunofluorescence. Cytolytic activity is influenced by factors such as TMB, particularly in MSI-H/dMMR cancers, which respond better to ICIs. Cytolytic activity is also closely linked to antigen presentation and the TME, including immune and stromal components. Clinically, cytolytic activity represents a valuable biomarker, with a high activity generally linked with better survival, except in glioblastoma, where a high cytolytic activity is associated with poorer outcomes [135].

Tertiary lymphoid structures (TLSs) and tumor-infiltrating B cells have been identified as potential biomarkers and therapeutic targets for the ICI therapy of various solid cancers, including melanoma [136], renal carcinoma [137], soft-tissue sarcoma [138], and lung cancer [139]. TLS can be present at different locations within tumor tissues, including in the stroma and parenchyma of infiltrative margins of tumoral tissues. These structures consist of follicular dendritic cells, follicular T cells, T cells, B cells, and fibroblasts and contribute to the regulation of the local immune response, playing a critical part in modulating the TME. It was found that TLS-related genes (such as CXCL13) and tumor-infiltrating B cells are strong predictors for ICIs response in patients with lung cancer [140]. Also, TLSs were reported to harbor atypical memory B cells (FCRL4+FCRL5+ B cells) in the TME of lung cancer patients that were correlated with ICIs’ response to therapy [132].

TLSs were demonstrated to induce the activation of effector immune cells and antigen-presenting cells, allowing them to circumvent the classical migration route between tumor tissues and draining lymph nodes. This enables a faster and localized immune response near the tumor, contributing to the amplification of immunity against cancer cells. However, the key factors driving TLSs’ formation, and their contribution to antitumor efficacy, remain unclear, largely due to the heterogenous functional status of TLSs [127]. Recent studies suggest that intra-tumoral TLSs may serve as biomarkers for ICIs’ response and may provide new avenues for personalized therapy in hepatocellular carcinoma, head and neck squamous cell carcinoma [139], and breast cancer [141].

Tumor-associated macrophages (TAMs) have also been identified as biomarkers for therapeutic outcomes [142], including ICIs response [143,144]. TAMs’ repolarization from M2 to M1 [145], an approach that could affect the therapeutic response, or the application of M1 TAMs as drug delivery vectors, has received significant attention [146]. Recently, Coulton et al. [147] published a comprehensive single-cell RNA-seq atlas of TAMs in cancer, capturing the full extent of their phenotypic diversity. The composition of TAMs was found to correlate with tumor phenotype, with distinct TAM subsets observed in primary versus metastatic cancers. Notably, the study identified two TAM subsets linked to the activation of T cells. Furthermore, an analysis of TAM profiles in a large group of ICI treated-patients (CPI1000+) revealed several TAM subpopulations as biomarkers for therapeutic response, such as one characterized by the upregulation of collagen-related genes. A recent report indicated that collagen-producing macrophages restrict CD8+ T-cell activity [148]. The TME may promote an increase in TAM levels by enhancing their differentiation into myofibroblasts, potentially contributing to resistance to ICI treatment. Further investigation of this specific TAM subset will be important in future studies [147].

Regulatory T cells (Tregs), defined by the co-expression of CD4, CD25, and FoxP3, play a key role in immune suppression in the TME through IL-10 secretion [149]. Furthermore, IL-35 secretion by Tregs inhibits the proliferation of T cells and acts synergically with IL-10 to induce an exhaustion phenotype in CD8+ T cells [150]. Treg recruitment into the TME is accomplished via CCR4/CCL22 or CCR4/CCL17 pathways after interactions between CCR4+ Tregs and CCL22/CCL17 secreted by TAMs and tumor cells [151]; as such, CCR4+ Tregs abundantly infiltrate the TME and are considered the most immunosuppressive subset of Tregs. The expression of PD-1/PD-L1 has not yet been revealed to be a reliable predictor of ICIs response in renal cell carcinoma, and its potential as a biomarker remains to be determined. In a study by Denize et al. [152], an unfavorable outcome in patients with metastatic renal cell carcinoma treated with nivolumab was correlated with the upregulation of PD-1 on Tregs combined with a low PD-1 expression on non-terminally exhausted CD8^+^ effector T cells. These findings indicate that Tregs expressing PD-1 may indicate resistance to ICIs, whereas the expression of PD-1 on non-terminally exhausted CD8+ T cells may play a protective role against resistance [152].

Cancer-associated fibroblasts (CAFs) are an important part of the TME, mainly in solid cancers. In malignancies such as breast and pancreatic carcinomas, often CAFs constitute the majority of stromal cells, their abundance being correlated with a poor outcome. CAFs are very diverse in terms of origin and roles, originating from resident tissue fibroblasts (reprogrammed by tumor cells), mesenchymal cells recruited into the TME from the bone marrow, or endothelial and mesothelial cells [153], and they are known to induce extracellular matrix (ECM) degradation and angiogenesis while also promoting tumor growth and immune evasion [154]. Their activation within the TME is driven by tumor-secreted factors such as transforming growth factor-β (TGF-β), platelet-derived growth factor (PDGF), hepatocyte growth factor (HGF), and epidermal growth factor (EGF) [155]. HGF is also responsible for the conversion and reprogramming of normal fibroblasts into CAFs [156].

CAFs were demonstrated to significantly impact tumor growth and drug resistance [157]. In patients with gastric tumors, Song et al. [158] reported a subset of CAFs known as antigen-presenting CAFs (apCAFs), expressing a high level of HLA II. They were primarily located near tertiary lymphoid structures, which mainly consist of T and B cells [158]. apCAFs were demonstrated to promote T-cell activation, increasing their cytotoxicity and proliferation, as well as macrophage polarization toward the M1 phenotype. Subsequently, M1 macrophages contribute to the expansion of apCAFs, establishing a self-reinforcing loop that increases antitumor immune activity. Notably, the study found that tumors from patients who responded to ICIs in different cancer types exhibited higher levels of apCAF within the TME. This suggests that the presence of apCAFs could serve as a potential biomarker for predicting ICIs’ responses [158].

In contrast, Zhang et al. recently described a subset of CAFs in prostate cancer, termed FerroCAFs, which accumulate iron and contribute to an immunosuppressive TME. These FerroCAFs secrete myeloid cell-associated proteins like CCL2, CSF1, and CXCL1, attracting immunosuppressive myeloid cells that hinder the body’s anti-tumor response. The presence of FerroCAFs correlates with poorer clinical outcomes in prostate cancer patients. Mechanistically, the study found that FerroCAFs accumulate iron through Heme Oxygenase 1 (Hmox1)-mediated heme degradation. This intracellular iron activates Kdm6b, an iron-dependent epigenetic enzyme, leading to chromatin remodeling and the increased expression of genes encoding myeloid cell-associated proteins. Targeting the Hmox1/iron/Kdm6b signaling pathway in FerroCAFs was shown to enhance anti-tumor immunity and suppress tumor growth in experimental models. Furthermore, the study reported the presence of FerroCAFs not only in prostate cancer but also in human lung and ovarian cancers, suggesting a broader role for these cells in tumor immunosuppression across different types of cancer. The identification of a conserved cell surface marker, the poliovirus receptor, on FerroCAFs may offer a potential target for therapeutic interventions aimed at boosting anti-tumor immunity [159].

### 4.4. Circulating Biomarkers

Liquid biopsy provides insights into tumors’ biology and resistance, reflecting the tumors’ environment. Circulating tumor cells, circulating tumor-derived nucleic acids (ctDNA), circulating cell-free DNA (cfDNA), microRNAs, exosomes, immune biomarkers, and tumor-educated platelets can be isolated from blood and other body fluids, such as saliva, urine, sperm, tears, and cerebrospinal fluid. Liquid biopsy enables real-time, non-invasive monitoring for personalized care. However, technical, economic, and standardization challenges hinder its clinical use. Overcoming these barriers through validation and standardization, along with AI and the integration of multi-omics, could revolutionize cancer care.

Circulating tumor cells can be more challenging to isolate than circulating tumor *DNA* derived from tumor cells. However, the analysis of CTCs allows functional assays and the assessment of CTC-derived DNA, RNA, and proteins [160]. The genetic makeup of CTCs may provide information about heterogeneity, mutations, and potential resistance to treatments [161]. Circulating tumor DNA can be tracked and measured qualitatively (methylation) and quantitatively and used to indicate the efficiency of ICI therapy in different types of cancers. In NSCLC patients, the ctDNA modification and level can be used to predict treatment efficacy.

Similarly, circulating immune cells were also exploited for the prediction of ICI therapy response. For instance, baseline levels of classical monocytes (CD14^+^CD16^−^) were found to be elevated in responders compared to non-responders with metastatic renal cell carcinoma [162]. In NSCLC patients who responded to ICIs, a significant reduction in both CD4^+^CD25^+^CD127^lo^FoxP3^+^ regulatory T cells and their PD-1^+^ counterparts were observed [163]. Additionally, baseline T-cell activation has emerged as a promising non-invasive biomarker for distinguishing responders from non-responders in NSCLC. Notably, an increased IL-2 production by CD8^+^ T cells following ex vivo activation at baseline was associated with favorable responses to ICI therapy [164]. Circulating immunosuppressive cells were identified as biomarkers for ICI therapy response [165]. Koh et al. [165] demonstrated that in patients with advanced NSCLC undergoing anti-PD-1 therapy (pembrolizumab or nivolumab), elevated circulating levels of Treg cells (CD25^+^FOXP3^+^CD4^+^) one week after onset of therapy were strongly associated with an improved PFS and OS.

The neutrophil-to-lymphocyte ratio (NLR) can indicate the response to ICI therapy. In a recent comprehensive review, Su et al. [166] reported that an elevated pre-treatment NLR in patients with advanced cancers was strongly correlated with a poor prognosis and limited ICIs response. A cut-off value of four for NLR was associated with prognosis and could represent a potential prognostic biomarker [166]. For clinical implementation, further validation through large-scale prospective studies to confirm its reliability and to elucidate the underlying biological mechanisms is required.

The plasma level of soluble PD-L1 (sPD-L1) has been identified as a potential biomarker in advanced cancers treated with ICIs. sPD-L1 can impair effector T-cell function and act as a decoy for anti-PD-L1 antibodies, reducing treatment’s efficacy [167]. Increased levels of sPD-L1 indicated a poorer outcome in different types of cancers, including esophageal, melanoma, lung, and gastric cancers. Additionally, dynamic changes in sPD-L1 levels may reflect tumors’ adaptation to immune activation, offering a tool for monitoring response and stratifying patients during immunotherapy [168].

### 4.5. Transcriptome Signatures

Transcriptomic signatures—patterns of gene expression derived from RNA sequencing—are gaining prominence as predictive biomarkers for ICI therapy. Transcriptomic signatures work by capturing the activity of immune-related genes, the tumor microenvironment’s composition, and the expression of checkpoint ligands and receptors. These profiles may indicate the presence of “hot” tumors—characterized by immune infiltration and activation—or “cold” tumors lacking immune engagement. A comprehensive meta-analysis by Yang et al. (2024) involving over 3000 patients treated with ICIs identified multiple immune activation markers—including IFNG and PDCD1 expression—and macrophage–T-cell interactions as central to therapeutic efficacy. They proposed an eight-signature model validated across diverse types of cancer [169]. A recent study by Xie et al. (2025) has further reinforced the value of transcriptomic and immune–molecular predictors. In colon cancer, a six-gene Metabolism and Immune-Related Prognostic Score (MIRPS)—comprising CD36, PCOLCE2, SCG2, CALB2, STC2, and CLDN23—was developed and validated across two independent cohorts [170]. The MIRPS was linked to a higher tumor mutational burden, immune escape characteristics, and ICI responsiveness, offering a dual prognostic and predictive utility. Xia et al. (2025) demonstrated that conserved immune signatures, particularly those involving B-cell markers, predict ICI efficacy across 20 tumor types, highlighting the importance of immune cell co-infiltration, and not just individual cell type abundance, in shaping therapy outcomes [171].

Transcriptomic profiling holds significant promise for identifying patients who are likely to respond to ICI therapies. However, translating transcriptomic data into robust and clinically useful biomarkers remains challenging due to a combination of biological, technical, and regulatory barriers. One of the major biological limitations is tumors’ heterogeneity. Gene expression profiles can vary widely, not only between different patients (inter-tumoral heterogeneity), but also within different regions of the same tumor (intra-tumoral heterogeneity). This variability complicates the development of consistent and reliable transcriptomic signatures. Moreover, the TME adds another layer of complexity. Much of the immune-related transcriptomic signal arises not from tumor cells themselves but from surrounding immune and stromal cells, such as T cells, macrophages, and fibroblasts. These mixed signals can obscure the specific contributions of tumor-intrinsic pathways to ICI responsiveness. Another key biological challenge is the dynamic nature of tumor–immune interactions [171]. Transcriptomic snapshots typically reflect a single time point, often before treatment. However, the response to ICIs involves temporal processes like T-cell activation, recruitment, and reversal of exhaustion, which evolve over time and are not captured in baseline gene expression data. Thus, a single pre-treatment biopsy may not adequately reflect the full immunological landscape necessary to predict a response.

Technically, transcriptomic analyses are sensitive to sample quality. Variability in RNA integrity, processing techniques (e.g., fresh-frozen vs. formalin-fixed samples), and sequencing platforms can introduce significant batch effects, undermining reproducibility across studies. Moreover, there is no standardized or universally accepted transcriptomic signature for ICIs response. While many models exist, such as IFNG expression, CD8 T-cell scores, and PD-1 pathway gene sets, their predictive power is often limited to specific cancer types or clinical settings. Bulk RNA sequencing remains the most widely used method due to its scalability, but it lacks the resolution to distinguish cell-type-specific gene expression, which is critical in the immune context. In contrast, single-cell RNA-seq provides much finer detail but is costly, technically complex, and not yet feasible for routine clinical use [172].

Computationally, integrating transcriptomic data with other biomarkers like TMB, spatial profiling, or proteomics increases predictive accuracy but also adds interpretive and analytical complexity [173]. Many models suffer from overfitting during development and fail to generalize across independent datasets, which hinders clinical adoption. Additionally, many of the most accurate models are built using machine learning approaches that lack transparency, making biological validation difficult and limiting clinician confidence. Regulatory and economic issues also pose barriers. Despite the growing body of evidence supporting transcriptomic predictors, few have made it through the regulatory pipeline due to the lack of large-scale validation studies and standardized pipelines. The costs associated with RNA sequencing, particularly longitudinal sampling or single-cell techniques, remain high, restricting its widespread use.

### 4.6. Metabolic Products

Several metabolites have emerged as promising biomarkers for predicting and monitoring responses to cancer immunotherapy (Table 2). Among them, the kynurenine/tryptophan ratio has been widely studied, as it reflects the activity of indoleamine 2,3-dioxygenase (IDO1), an enzyme implicated in immune suppression through tryptophan depletion and kynurenine accumulation. Elevated levels are frequently associated with resistance to ICIs [174].

Lactate dehydrogenase has been identified as a biomarker for predicting ICI responses in patients with advanced NSCLC [175,176]. A reduction in LDH levels exceeding 20% was significantly associated with improved radiological responses [175]. Conversely, elevated baseline LDH levels were associated with a decreased overall survival (OS) and progression-free survival (PFS). These findings emphasize the prognostic and potentially therapeutic relevance of lactate metabolism in the context of lung cancer treatment. Supporting this, a recent study found that the expression of lactate metabolism-related genes in tumor cells was linked to patient survival, immune cell infiltration, and ICIs response in lung adenocarcinoma. Notably, a high lactate metabolism risk score was associated with the limited efficacy of ICIs [177].

Arginine, essential for T-cell activation, when depleted by tumor-expressed arginase, can also contribute to poor immunotherapeutic outcomes [178]. On the other hand, short-chain fatty acids (SCFAs) like butyrate and propionate, derived from gut microbiota, have been linked to an enhanced T-cell function and improved response to ICIs [179]. Alterations in lipid metabolism, reflected in acylcarnitine profiles, and elevated polyamines such as spermidine and spermine, are also associated with immune suppression and poor outcomes [180]. Glutamine metabolism serves a dual function by supporting the bioenergetic and biosynthetic demands of both tumor and immune cells. Dysregulation of this metabolic pathway can significantly impact the efficacy of immunotherapeutic interventions [181].

**Table 2 ijms-26-06177-t002:** Metabolites with biomarker value in ICI therapy response.

Metabolite	Biomarker Role	Cancer Type	Impact for Immunotherapy	References
Kynurenine/IDO1	Immune suppression marker	Metastatic renal cell carcinoma Acute myeloid leukemia Glioblastoma Hepatocellular carcinoma.	Predicts poor response	[182,183,184,185]
Lactate	T-cell suppression in TME	Pan-cancer Pancreatic cancer	Associated with resistance	[186,187,188]
Arginine	T-cell proliferation and activation	Liver cancer	Low levels = reduced efficacy	[189]
SCFAs (e.g., butyrate)	Immune modulation via gut microbiome	Solid tumor	Correlates with better outcomes	[190]
Acylcarnitines	Lipid metabolism dysregulation	Acute myeloid leukemia Hepatocellular carcinoma	Linked to immune dysfunction	[191,192]
Polyamines	Tumor-promoting, immunosuppressive	Colorectal cancer	Elevated in non-responders	[193]
Glutamine	Supports tumor and T-cell metabolism	Lung adenocarcinoma	Metabolic imbalance affects response	[194]

### 4.7. Microbiome

The human microbiome, particularly the gut microbiota, has a fundamental role not only in maintaining systemic homeostasis but also influencing cancer’s initiation and progression and the host immune response against tumors [195,196]. A healthy gut microbiome, often referred to as a state of eubiosis, is characterized by a high microbial diversity and the dominance of beneficial microbes such as Firmicutes [197]. These microbes produce metabolites like short-chain fatty acids, which enrich epithelial barriers’ function and promote regulatory T-cell activity, crucial for balanced immune responses. In contrast, dysbiosis—a disrupted microbial balance—can lead to an overrepresentation of pathogenic microbes such as Proteobacteria, which are associated with systemic inflammation and an impaired treatment efficacy [198]. Microbial communities can modulate both local and systemic immunity through the production of immunologically active metabolites, including short-chain fatty acids, indoles, secondary bile acids, and polyamines [199]. These microbial signals influence key processes, including T-cell differentiation, dendritic cell activation, and mucosal barrier integrity. Dysbiosis—an imbalance in the microbiota—has been linked to chronic inflammation, DNA damage, oxidative stress, and disruption of epithelial barriers, all of which contribute to carcinogenesis in tissues such as the colon, liver, pancreas, and stomach [200,201]. Specific pathogens, like *Helicobacter pylori* in gastric cancer or *Fusobacterium nucleatum* in colorectal cancer, have been directly implicated in tumorigenesis through mechanisms that include immune evasion, pro-inflammatory signaling, and the modulation of tumor-promoting microenvironments [202,203].

Furthermore, the microbiome can influence tumor-associated macrophage polarization, the balance between effector and regulatory T cells, and the recruitment of myeloid-derived suppressor cells—factors that collectively shape tumor immunity. Emerging evidence suggests that microbial signatures within the gut, tumor microenvironment, and even distant organs may serve as biomarkers for cancer’s progression or therapeutic responsiveness. As such, the microbiome represents a key node in the complex network connecting host immunity and cancer biology, offering novel targets for prevention, diagnosis, and therapy [204].

The human microbiome—particularly the gut microbiota—is considered a pivotal modulator of host immunity and a critical determinant of cancer immunotherapy outcomes [205]. Clinical evidence is beginning to validate these associations. For example, in a prospective study on patients with pancreatic ductal adenocarcinoma (PDAC), those with higher levels of Firmicutes and lower levels of Proteobacteria showed improved responses to combination treatments, including gemcitabine, nab-paclitaxel, and the immune checkpoint inhibitor durvalumab. In contrast, patients with high Proteobacteria and reduced diversity had worse outcomes, suggesting that specific microbial profiles may predict or even influence therapy’s success [206]. Groundbreaking studies over the past decade have demonstrated that the gut microbiome not only influences the baseline immune tone but also modulates the therapeutic efficacy and toxicity profile of ICIs, including anti-PD-1, anti-PD-L1, and anti-CTLA-4 antibodies [207]. Patients with a diverse and compositionally favorable gut microbiome tend to exhibit superior responses to ICIs, characterized by prolonged progression-free and overall survival [208,209,210].

Specific commensal bacteria such as *Akkermansia muciniphila*, *Faecalibacterium prausnitzii*, *Bifidobacterium longum*, *Ruminococcus* spp., and *Alistipes* have been repeatedly associated with improved immunotherapy responses [211]. These microbes may enhance antitumor immunity through several mechanisms: guiding the maturation and function of antigen-presenting cells, increasing the infiltration and activity of effector T cells within the tumor microenvironment, modulating the expression of co-stimulatory molecules, and producing metabolites like SCFAs, inosine, and tryptophan derivatives that can directly influence immune signaling pathways.

In contrast, dysbiosis—characterized by a reduced microbial diversity and an overrepresentation of pathobionts such as *Enterococcus faecalis* or *Escherichia coli*—has been linked to immunotherapy resistance, poor treatment outcomes, and a heightened susceptibility to immune-related adverse events, including colitis and pneumonitis [212,213,214,215]. Factors contributing to dysbiosis include prior exposure to antibiotics, chemotherapy, radiotherapy, high-fat or low-fiber diets, chronic inflammation, and comorbid conditions such as obesity or metabolic syndrome. Alarmingly, the use of broad-spectrum antibiotics shortly before or during ICI therapy has been shown in multiple cohorts to significantly impair treatment’s efficacy, underlining the fragile balance between microbial composition and immune competence.

Beyond taxonomic composition, the functional capacity of the microbiome—its collective metabolic output and gene expression—appears equally important. Microbial-derived metabolites such as butyrate, acetate, propionate, polyamines, and secondary bile acids have profound effects on regulatory T-cell function, dendritic cell activation, and epithelial barrier integrity. In particular, SCFAs play a dual role: at physiological levels, they promote mucosal immune homeostasis and anti-inflammatory responses, while under certain conditions, they may prime systemic immunity and enhance the efficacy of ICIs. Furthermore, the microbiome-driven modulation of systemic cytokine profiles—such as increased IL-12, IFN-γ, and TNF-α—has been observed in responders to checkpoint blockade, suggesting a role in bridging gut and tumor immune environments [216,217].

These insights have catalyzed a wave of translational research aimed at therapeutically manipulating the microbiome to optimize cancer immunotherapy. Fecal microbiota transplantation (FMT) from responders to non-responders has shown encouraging results in early-phase clinical trials, restoring sensitivity to the PD-1 blockade in refractory melanoma [218,219]. Similarly, defined consortia of live bacteria, or next-generation probiotics, are being developed as adjunctive therapies to ICIs. Dietary strategies, such as increasing dietary fiber intake or consuming fermented foods, are being tested as low-risk, accessible interventions to support a beneficial gut ecosystem. Postbiotics—the use of microbial metabolites or inactivated microbes—offer another promising avenue, especially when live bacterial therapies are not feasible. In parallel, efforts are underway to identify microbial biomarkers that can predict immunotherapy responsiveness, paving the way for personalized treatment regimens that incorporate microbiome profiling as a standard clinical tool.

## 5. Discussion

Future directions in cancer immunotherapy are centered on the development of next-generation strategies that enhance its efficacy, precision, and accessibility. A major direction involves the refinement of combination therapies—merging ICIs with cancer vaccines, targeted therapies, or metabolic modulators—to overcome resistance and expand the therapeutic response across tumor types [220]. Innovations in adoptive cell therapy (ACT), including the engineering of more potent and persistent CAR-T and CAR-NK cells, in addition to the application of allogeneic or induced pluripotent stem cell (iPSC)-derived immune cells, are expected to broaden clinical applications while reducing costs and manufacturing time [76,221,222]. Personalized neoantigen-based vaccines, driven by real-time genomic sequencing and artificial intelligence, are being integrated into treatment plans to prime tumor-specific immunity [66]. Cytokine-based therapies are also evolving, with engineered cytokine variants and orthogonal receptor–cytokine systems designed to enhance antitumor immunity while minimizing toxicity [223]. Furthermore, overcoming immune exclusion and resistance in “cold” tumors remains a central challenge; future strategies may include epigenetic modulators, oncolytic viruses, and microbiome interventions to remodel the TME and facilitate immune infiltration [224,225]. In addition, TIL therapy is also a promising avenue in cancer immunotherapy. The recent FDA approval of lifileucel (Amtagvi) in 2024 for advanced melanoma underscores the potential of TIL therapy in treating solid tumors. Ongoing research aims to expand TIL therapy to other cancers, such as NSCLC and breast cancer, by enhancing TILs’ persistence and tumor specificity through genetic modifications and combination strategies. Efforts are also underway to streamline manufacturing processes, making TIL therapy more accessible and cost-effective for a broader patient population [226]. Moreover, the convergence of immunotherapy with systems biology, machine learning, and nanotechnology is poised to usher in an era of highly tailored and effective treatments for a broader patient population.

Looking forward, the future of immunotherapy lies in advancing toward more personalized, dynamic, and combinatorial strategies. Advancing immunotherapy relies on overcoming resistance, enhancing biomarkers’ accuracy, and reshaping the tumor microenvironment—key steps toward expanding its effectiveness and reach across diverse cancer types. Novel approaches such as neoantigen-based vaccines, bispecific antibodies, CAR-T-cell therapies, and oncolytic viruses are expanding the therapeutic options beyond conventional ICIs [227]. The integration of multi-omics data—spanning genomics, transcriptomics, proteomics, and microbiomics—enhances our ability to understand tumor–immune interactions and tailor treatments to individual patient profiles.

Among emerging strategies, novel combination strategies are being actively explored to improve the therapeutic response to ICIs. One particularly promising approach involves pairing single-agent ICIs with personalized, neoantigen-based tumor vaccines designed to prime tumor-specific T-cell responses prior to ICIs’ administration. This strategy aims to stimulate a robust and highly targeted immune response, effectively “training” the immune system to recognize and eliminate cancer cells. By priming the patient’s immune system in advance, this combination has the potential to significantly enhance the efficacy of ICIs. Clinical studies have shown that such combinations not only expand the repertoire of tumor-specific T cells but also help convert immunologically “cold” tumors—those with a low immune infiltration—into “hot” tumors, which are more responsive to immunotherapy [228,229].

Combining PI3K inhibitors with ICIs has emerged as another promising strategy to improve immunotherapy outcomes [230]. Recent studies have demonstrated that the novel pan-PI3K inhibitor KTC1101 not only suppresses tumor growth but also improves the therapeutic response to anti-PD-1 treatment across multiple preclinical mouse models [231]. Similarly, in murine models of triple-negative breast cancer, the dual PI3K/mTOR inhibitor gedatolisib has been shown to significantly enhance ICIs’ efficacy, resulting in robust tumor growth inhibition and the activation of anti-tumor immune responses [232]. Furthermore, in patients with metastatic head and neck cancer, the combination of the PI3K inhibitor alpelisib with immunotherapy has demonstrated encouraging results, underscoring the potential of this combinatorial approach [233].

Even though immunotherapy is playing an increasingly important role in cancer treatment, a significant number of patients do not respond to treatment. Consequently, ongoing research is heavily focused on identifying predictive biomarkers that can support the selection of patients most likely to benefit from immunotherapy. One of the major challenges in discovering such biomarkers lies in the complex and heterogenous nature of cancer. The wide range of pathogenic mechanisms and the biological heterogeneity across different types of cancers makes it difficult to identify reliable biomarkers of cancer response [234].

Dynamic changes in the TME further complicate the interpretation of static biopsy samples. The spatial and temporal evolution of immune cell infiltration, stromal interactions, and cytokine signaling can lead to discordance between biomarker expression at biopsy and actual therapeutic response [235].

Assessing tumor tissue biomarkers in clinical practice faces several key limitations: (i) the limited availability and accessibility of tumor biopsies, as current companion diagnostic assays rely on tissue samples; (ii) tumor heterogeneity, which can cause a significant variability in evaluating markers like PD-L1 expression, TMB, or MSI depending on the biopsy section analyzed; and (iii) the presence of additional molecular mechanisms influencing the response to ICIs that may not be captured through standard tissue-based assessments [236].

Discrepancies in testing platforms, scoring thresholds, and pathologist interpretation introduce a variability that can undermine the reliability of predictive biomarkers in real-world settings. In line with this, emerging liquid biopsy approaches, such as circulating tumor DNA (ctDNA), exosomes, and immune cell profiling, offer promising non-invasive alternatives. However, these technologies face their own challenges, including a low sensitivity for early-stage cancers, signal dilution in highly heterogeneous tumors, and the need for rigorous validation across cancer types and treatment contexts [237].

Experts are proposing a novel strategy to overcome the limitations of single biomarkers by advancing the use of combined multiple biomarkers [238] to better reflect disease mechanisms, thereby offering a better stratification of patients and better outcomes. This multiple or combined marker approach could support the advancement of personalized treatments, enabling more informed treatment decisions through a comprehensive understanding of each patient’s unique biological profile. For example, mutations in ARID1A and elevated expression of the chemokine CXCL13 have been associated with an improved response to ICI therapy in patients with metastatic urothelial cancer [239]. Similarly, high levels of TMB, a T cell inflamed gene expression profile (GEP), and PD-L1 expression have been shown to collectively predict a greater likelihood of response to pembrolizumab across various cancer types [240]. Advances in technologies such as machine learning and artificial intelligence are enhancing the comprehensive analysis of multiple biomarkers. Given the complex and multifactorial nature of tumor–immune interactions, predictive models that integrate a wide range of biomarkers may provide a more practical and effective approach for future clinical applications.

Lastly, the interplay of host-related factors—such as gut microbiome composition, systemic inflammation, and genetic polymorphisms—add layers of complexity to patient stratification [241].

The microbiome holds promise for boosting immunotherapy, but its clinical use faces major challenges. Each person’s microbiome is shaped by genetics, diet, lifestyle, and more, making it hard to define a universal “beneficial” profile. Individual variability means responses to microbiome-based interventions can differ widely. Genetic factors like HLA types and polymorphisms in innate immunity genes can influence microbial colonization patterns and modulate how the immune system interprets microbial signals [242]. Ignoring these host–microbe interactions risks oversimplifying the microbiome’s role in therapy.

The microbiome’s impact on immunotherapy varies by type of cancer. In skin cancers, microbes like *Bifidobacterium* and *Lactobacillus* may boost checkpoint inhibitor responses by enhancing T-cell activity [225]. While FMT and probiotics show promise in melanoma, applying these strategies to other cancers is difficult due to differing tumor environments and host–microbiome dynamics [243]. The microbiome influences immunity through a web of direct and indirect pathways involving metabolites, cytokines, barrier function, and antigen presentation [244]. In addition, there is no universally accepted pipeline for microbiome analysis. Differences in sample types (e.g., stool vs. mucosal), DNA extraction methods, sequencing platforms, and bioinformatics tools lead to inconsistent and hard-to-replicate findings. These technical discrepancies hinder inter-study comparisons and meta-analyses, underscoring the urgent need for methodological consistency in cancer-related microbiome studies [245]. Understanding how the microbiome influences tumor immunity across cancer types is challenging. Its variability over time complicates identifying consistent biomarkers. Most findings are correlative, not causal, and proving direct effects requires complex models or interventions like FMT [246].

Technical limitations further impede progress. Traditional sequencing methods can be affected by contamination, especially in low-biomass tumor samples, leading to false-positive microbial identifications. To address this, new computational tools like PRISM have been developed. PRISM enhances microbial detection accuracy by decontaminating sequencing data, enabling a more reliable identification of tumor-associated microbes and their potential clinical significance [247].

Adding complexity, many preclinical findings rely heavily on mouse models, which often fail to translate to human settings. Differences in microbiome composition, immune system architecture, and tumor biology between mice and humans can distort predictions about therapeutic outcomes. A systematic review of drug delivery in animal cancer models emphasized this translational gap and the need for more human-relevant systems [248].

Confounding factors like antibiotics, prior treatments, diet, and stress can alter the microbiome and skew study results. Antibiotics, in particular, can disrupt beneficial microbes critical for immunotherapy efficacy. Separating the effects of antibiotics from intrinsic microbiome–tumor interactions in clinical trials is thus a major methodological hurdle [206]. Despite promising preclinical data, large, controlled clinical trials on microbiome-targeted interventions in immunotherapy are scarce. Most evidence comes from small or retrospective studies. Ethical concerns around consent and data privacy, along with logistical hurdles in sample handling and biobanking, complicate large-scale research. A key unmet need remains the identification of reliable microbial biomarkers for patient stratification [249]. Moreover, the lack of a clear, validated microbiome-based diagnostic test makes it difficult to guide clinical decision-making. Microbiome interventions like FMT, probiotics, and diet show promise but are often unpredictable and hard to sustain. FMT lacks standardization; probiotic effects are strain-specific, and long-term microbiome shifts are uncertain. Regulatory and ethical issues—such as safety, donor screening, and international trial discrepancies—further complicate their clinical use.

We acknowledge that achieving an effective patient stratification in immuno-oncology is hindered by a series of interrelated challenges spanning biological, computational, and clinical domains (Figure 2). The tumor microenvironment’s complexity, the variability in patient responses, and the lack of reliable predictive biomarkers remain major obstacles in immunotherapy. Similarly, the microbiome presents difficulties due to interindividual variation, confounding factors, and the ongoing struggle to establish causality over correlation. On the computational side, AI-driven models face barriers related to data integration, models’ interpretability, and regulatory oversight, while transcriptomic approaches are limited by temporal variation and the need for single-cell resolution. These multifaceted issues collectively complicate the development of precision oncology strategies and highlight the need for integrative, standardized, and longitudinal solutions.

## 6. Future Directions

Ongoing research aims to identify new predictive biomarkers that evaluate both the tumor and the host immune environment. These efforts include studies on the TME, mutational landscapes, transcription factors, microRNAs, the microbiome, the neutrophil-to-lymphocyte ratio, and soluble biomarkers from liquid biopsies. While these biomarkers are designed to reflect the immunogenicity of both tumor and host, none have yet been validated for clinical use. In parallel, the development of multi-parametric, -omics technologies, such as multiplexed immunohistochemistry (IHC), immunofluorescence, single-cell transcriptomics, and mass spectrometry-based quantitative and spatial proteomics, is driving progress in this field. These cutting-edge platforms are expected to facilitate the discovery of more robust biomarkers and help overcome current challenges in immuno-oncology treatment [250,251] (Table 3).

The scarcity of validated biomarkers for predicting immunotherapy outcomes, combined with the variability in patient responses to ICIs and the critical role of the TME, presents a major challenge in cancer treatment. To address this, integrating machine learning (ML) and deep learning models with multi-omics and clinical data is now widely acknowledged as essential for advancing the predictive accuracy in cancer immunotherapy. Artificial intelligence (AI) plays a key role by processing vast amounts of clinical, genomic, and biomarker data to identify determinants of ICIs’ efficacy [252]. Moreover, AI can integrate multi-omics with clinical parameters to enhance the prediction in both immunotherapy and targeted treatments, ultimately supporting more precise therapeutic decisions for cancer patients [253]. AI has already demonstrated a broad potential in oncology, from tumor detection to prognosis. This section focuses on the use of ML and deep neural networks (DNNs) for predicting immunotherapy outcomes.

Machine learning has significantly transformed oncology via the creation of predictive models that support timely cancer detection, the planning of personalized therapies, and improving prognosis assessments across different types of cancer. Large-scale datasets are analyzed using sophisticated algorithms, enabling the discovery of hidden signatures and associations often overlooked by conventional statistical approaches [254]. Key predictive features such as PD-L1 expression, tumor mutational burden (TMB), and microsatellite instability (MSI) have been widely investigated in this context [255]. Some ML models integrate these biomarkers and apply conventional algorithms to predict outcomes within individual cancer types, as well as across multiple cancer types. A recent example, Zhang et al. (2023), used a machine learning approach to analyze gene expression and clinical data from 672 cancer patients across four cohorts (*n* = 348, bladder cancer, *n* = 98, 20 tumor types, *n* = 45, gastric cancer and *n* = 181, kidney cancer), all treated with ICIs. They ranked and normalized gene expression values, then examined over 1000 immune-related genes to identify signatures linked to treatment response and survival [256]. A machine learning model (Bayesian regularization neural networks, BRNNs) was trained on 75% of the bladder cancer cohort data and tested on the remaining 25%, then validated in the other cohorts. The model identified 40 key differentially expressed genes (DEGs) that could effectively predict ICIs response and survival outcomes. The model outperformed traditional biomarkers like TMB and gene expression profiling (GEP) in predicting treatment response, achieving a higher accuracy (AUC scores) across all cohorts. It also successfully distinguished long-term survivors from short-term survivors, showing a significant predictive power in different types of tumors [257].

Beyond conventional machine learning approaches, deep learning has become increasingly prominent in forecasting responses to cancer immunotherapy. By building multi-layered neural networks, deep learning models are capable of automatically learning and extracting intricate data features, resulting in an enhanced predictive accuracy. Among these, convolutional neural networks (CNNs) have shown effectiveness in predicting ICI responses by analyzing complex patterns within medical images, thereby achieving high levels of precision in the prediction of treatment outcomes.

Finally, integrating microbiome diagnostics with immunogenomic data could help stratify patients into responders and non-responders more accurately, guide pre-treatment microbial optimization, and reduce irAE’s incidence through the targeted modulation of gut microbiota. Ultimately, the microbiome represents a powerful, underexplored axis of immune modulation with the potential to transform the landscape of precision medicine in oncology and beyond.

To summarize, immunotherapy has been recognized as a transformative approach in cancer treatment, leveraging the host’s defense mechanisms to recognize and eliminate tumor cells. However, its efficacy varies significantly among patients, underscoring the need for predictive tools to guide selection of therapy. This is where cancer biomarkers play a pivotal role. Biomarkers not only aid in identifying the patients most likely to benefit from specific immunotherapeutic agents but also help monitor the treatment response and potential resistance. The integration of biomarker analysis into immunotherapy strategies thus represents a critical step toward more personalized and effective cancer care.

## 7. Conclusions

This review underscores the complex interplay between tumors and the immune system, highlighting the pivotal role of the tumor microenvironment in shaping immunotherapeutic responses. We have detailed key immune evasion strategies, emerging therapeutic modalities—including checkpoint inhibitors, CAR-T cells, cancer vaccines, and cytokine therapies—and the critical role of biomarkers in guiding treatment decisions.

Understanding how tumors evade the immune system is still a key piece of the puzzle in pushing immuno-oncology forward. Research in the coming years will likely zero in on discovering new ways cancer cells slip past immune defenses and how to block them effectively. Tying these insights into next-generation treatments, like tailored cancer vaccines, T-cell modifications, or combined checkpoint inhibitors, could help us crack the code on resistance and obtain more lasting results. As our grasp of these evasive tactics grows, it will be central to making immunotherapy work better across a broader range of cancers.

Alongside this, finding dependable biomarkers to predict how patients will respond to immunotherapy is becoming more and more important. While considerable progress has been made, the field must continue to address biological heterogeneity, optimize biomarkers’ utility, and better understand immune toxicity. Thanks to advances in transcriptomics and metabolomics, scientists are identifying molecular patterns that hint at whether a treatment will work or not. These tools are shedding light on both tumor-specific traits and the immune system’s role in therapy outcomes. Interestingly, the gut microbiome has also stepped into the spotlight, certain bacterial makeups seemingly boosting or dampening responses to treatment. Bringing together multi-omics data with microbiome analysis is shaping up to be vital for crafting truly personalized, biomarker-guided cancer therapies.

## Figures and Tables

**Figure 1 ijms-26-06177-f001:**
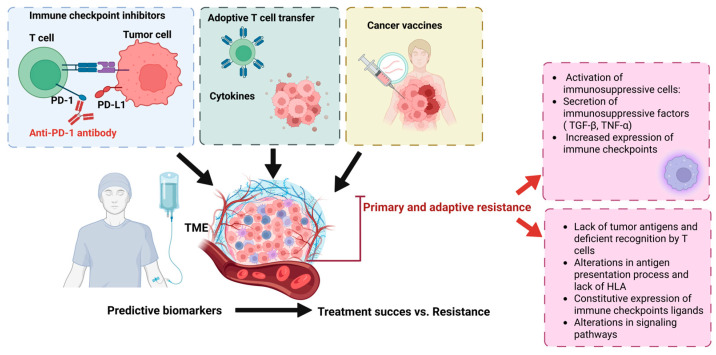
Key cancer immunotherapy strategies—ICIs, adoptive T-cell transfer, and cancer vaccines—interact closely with TME, which plays a central role in determining treatment success or resistance. ICIs function by blocking inhibitory signals like PD-1/PD-L1, thereby restoring T-cell activity. Adoptive T-cell transfer involves the introduction of tumor-reactive T cells supported by cytokines to enhance their function. Cancer vaccines aim to stimulate immune recognition of tumor specific-antigens. All these therapies converge on the TME, where immune responses are shaped by both tumor-intrinsic factors and the surrounding immune milieu. The diagram highlights two main resistance mechanisms: (1) immune-mediated, including the activation of suppressive cells and cytokines such as TGF-β; and (2) tumor-intrinsic, such as antigen loss, impaired antigen presentation, and constitutive expression of immune checkpoints.

**Figure 2 ijms-26-06177-f002:**
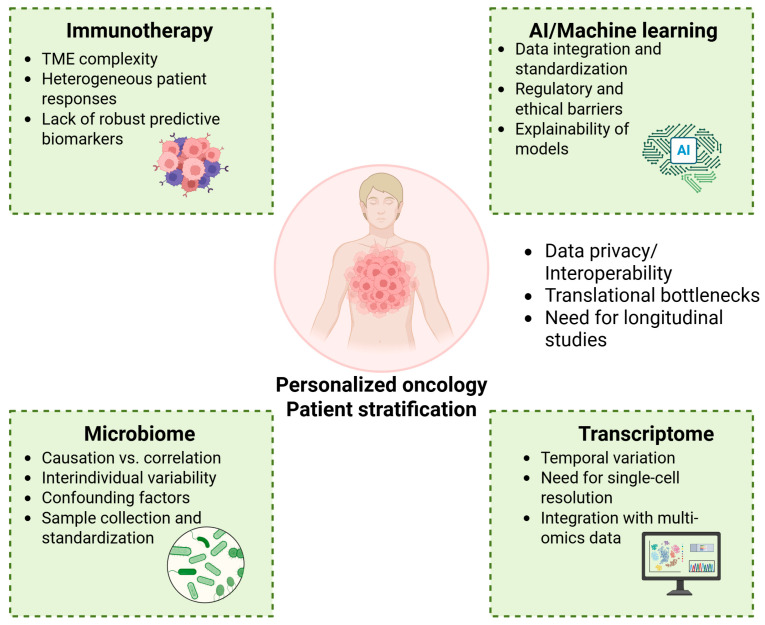
The multidimensional challenges at the convergence of immunotherapy, AI and machine learning, microbiome research, and transcriptomics, all aimed toward realizing the goal of personalized medicine.

**Table 3 ijms-26-06177-t003:** Innovations in multiplex tissue staining, especially through the integration of proteomic profiling, are transforming the landscape of spatial molecular diagnostics.

Multiplex Tissue Staining	Advantages	Limiting Factors	Perspectives
mIHC/IF technologies	In-depth research on functional cellular states and spatial dynamics related to cell-to-cell interactions within intricate tumor microenvironments (TMEs).	The workflow presents several challenges, including pre-analytical issues such as staining variability, as well as analytical complexity and difficulties in the interpretation and querying of post-analytical data.	The whole-slide AI-based “AstroPath” platform has identified predictive features in the pre-treatment of melanoma tissue samples that are associated with the response to anti–PD-1 therapy.
PhenoCycler-Fusion: single-cell phenotypes and spatial relationships via DNA-conjugated antibodies	Employs DNA-conjugated antibodies along with the cyclic addition and removal of complementary fluorescently labeled DNA probes to enable the simultaneous visualization of up to 60 markers in situ.	The approach is associated with high costs, primarily due to the use of antibodies and tagged DNA oligonucleotides. Additionally, the detection of low-abundance proteins often requires signals’ amplification, as the native signal is typically insufficient for reliable quantification.	Future advancements in this technology may involve the incorporation of nucleic acid labeling, enabling the simultaneous detection of nucleic acids. This could open new avenues for investigating causative genetic mutations and post-transcriptional modifications.
Single-cell RNA sequencing: TME gene expression and T-cell receptor sequencing	Powerful tool for thoroughly analyzing the tumor microenvironment (TME) to identify new and effective immunotherapies.	Single-cell RNA sequencing data is inherently noisy, making it difficult to establish clear correlations between genotype and phenotype due to technical limitations. These challenges are further amplified when analyzing cells derived from solid tumor tissues, where variability in tissue dissociation methods and cryopreservation conditions can significantly impact data’s quality and consistency.	Additional capabilities may include the inference of splice variants, chromosomal copy-number aberrations, and even the prediction of future cellular states. However, these advanced analyses require specialized expertise and careful interpretation to ensure accuracy and reliability.
Visium Spatial Gene Expression: barcoding transcriptomes across TMEs	It has facilitated the identification of different B-cell maturation states within tertiary lymphoid structures, utilizing Visium Spatial Gene Expression (SGE) in conjunction with pooled CRISPR screens.	Visium SGE is subject to several significant limitations, including suboptimal capture efficiency, restricted sequencing depth, and a high incidence of dropout events. These factors complicate the study of cell–cell interactions and the organization of higher-order tissue structures that influence immune responses. A key challenge is its lack of single-cell resolution, which restricts its ability to resolve fine-grained spatial details.	Integrating this approach with single-cell transcriptomics holds promise for overcoming current limitations in spatial resolution.
GeoMx: protein and transcriptome barcoding TMEs	It allows for a high-plex evaluation of transcripts (over 18,000 genes) and/or proteins (more than 100 proteins) within a single tissue sample. This technology is applicable to formalin-fixed, paraffin-embedded (FFPE), and fresh-frozen tissues. Interactive software facilitates collaboration, enabling the profiling of RNA transcripts and proteins according to the tissue’s spatial distribution.	The platform is relatively expensive, and whole-tissue analysis is more efficiently performed using alternatives like Visium or PhenoCycler-Fusion, which can profile the entire slide. Moreover, GeoMx lacks single-cell and subcellular resolution, limiting its utility for high-resolution spatial studies.	This approach has been employed to detect biomarkers associated with responses to bispecific antibody therapy in bone marrow biopsies. It has also been utilized to identify biomarkers linked to responses to cellular immunotherapies, including CAR T cell and transgenic T cell treatments.
Spatially resolved proteomics: mass spectrometry imaging and related technologies	This technique permits the in situ analysis of the spatial proteome, lipidome, glycome, and metabolome directly within tissue sections, without the need for specific staining or labeling, unlike many conventional visualization methods. It has been extensively applied in high-resolution studies of small molecular markers, including lipids, metabolites, and elemental species, as well as in the spatial characterization of drug compounds.	It lacks compatibility with the characterization of high-molecular-weight compounds.	It is anticipated that key advancements in MSImg will include the precise in situ analysis and visualization of novel biomarkers via single-cell spatial multi-omics, accelerated real-time examination of living tissues, and precision-guided surgery—all of which represent cutting-edge frontiers in the field.

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
