# Peer review of "Immuno-Oncology at the Crossroads: Confronting Challenges in the Quest for Effective Cancer Therapies"

_ijms, 2025, doi:10.3390/ijms26136177_

Round 1

Reviewer 1 Report

Comments and Suggestions for Authors

The manuscript "Immuno-Oncology at the Crossroads: Confronting Challenges in the Quest for Effective Cancer Therapies” provides a comprehensive overview of the mechanisms underlying tumor-immune system interactions and the therapeutic innovations emerging from this knowledge. This review provides a comprehensive summary, but the following aspects need to be addressed before the manuscript can be considered for publication.

  1. The manuscript alternates between “ICIs” and “immune checkpoint inhibitors” in different contexts. Use “ICIs” consistently throughout the manuscript.
  2. Modal verbs such as“can” must be followed by a verb prototype. There are grammar errors in the paper, please check the entire text.

A promising solution to this has been explored in combining single agent ICI with personalized neoantigen-based tumor-specific vaccines, a therapy that can primes the host’s defense mechanisms to the tumor beforehand, therefore enhancing ICI therapy response in patients.

  1. Research has shown that the occurrence of immune checkpoint inhibitor resistance(PD-1 inhibitors) in the treatment of tumors, especially head and neck cancer, is related to the activation of the PI3K pathway, and the combination of such immune checkpoint inhibitors and PI3K inhibitors is becoming increasingly widely studied. Please supplement relevant content to enhance the completeness of immunotherapy methods and improve the quality of the manuscript.

Reviewer 2 Report

Comments and Suggestions for Authors

I would like to sincerely commend the authors for their exceptional review, titled “Immuno-Oncology at the Crossroads: Confronting Challenges in the Quest for Effective Cancer Therapies.” This manuscript is both timely and comprehensive, offering an exceptionally well-structured analysis that captures the intricate complexities and dynamic advancements within the current landscape of immuno-oncology.

The review stands out due to several key strengths:

The manuscript skillfully integrates foundational immunological principles with the latest developments in the field, providing a comprehensive overview of the subject. It provides an extensive overview of topics such as immune checkpoint inhibitors, chimeric antigen receptor-based therapies, cancer vaccines, cytokine modulation, and novel therapeutic targets. Each section is thoroughly substantiated with up-to-date, high-quality references, including pivotal studies published, which enhance the manuscript’s credibility and authority.

Additionally, the authors explore innovative and emerging concepts crucial for advancing immuno-oncology. Their discussion of extrachromosomal DNA as a significant mechanism of resistance to therapy is particularly noteworthy. Additionally, they explore the application of spatial transcriptomics technologies, such as PhenoCycler-Fusion and Visium, in understanding tumor microenvironments. The exploration of tissue-resident memory T cells and the integration of AI-assisted models, notably AstroPath, provide a profound understanding of the direction in which the field is evolving, offering readers invaluable insights into future research and therapeutic strategies.

It is also critical that this review adeptly bridges the gap between basic science and clinical application. The authors have contextualized key concepts, including therapeutic resistance and biomarker development, such as PD-L1 expression, tumor mutational burden, tumor-infiltrating lymphocytes, and transcriptomic signatures, within the framework of real-world challenges that exist as a significant challenge. They extensively address issues such as variable patient responses to therapies and the logistical challenges associated with delivering advanced cell therapies, highlighting that the content is not only theoretically rich but also practically relevant for clinicians and researchers alike.

The manuscript's structure is exceptionally well-organized and maintains a logical flow throughout. Transitions between sections are flawless, enabling a coherent narrative that builds to a compelling conclusion. The figures and tables are thoughtfully designed, enhancing comprehension and providing visual support for complex concepts without causing redundancy.

The authors show an admirable balance between enthusiasm for innovations in the field and a critical perspective on the challenges that remain. They thoughtfully discuss issues such as immune-related adverse events, various resistance pathways, and the ethical considerations surrounding emerging areas, including microbiome modulation and access to advanced therapies. This balanced approach enriches the discussion and underscores the multifaceted nature of the field. Considering the comprehensive scope, scientific rigor, clarity of presentation, and substantial value this manuscript offers to the field of immuno-oncology, I have no critical concerns or necessary revisions. Hereby, I recommend accepting this review as it stands.

Congratulations to all authors for producing a highly impactful and scholarly review that will serve as a valuable reference for both researchers and clinicians navigating the rapidly evolving field of cancer immunotherapy.
